# Age-Graded Transitions and Turning Points in Polish Offenders’ Criminal Careers from the Standpoint of Life Course Theory

**DOI:** 10.3390/ijerph18116010

**Published:** 2021-06-03

**Authors:** Krzysztof Pękala, Andrzej Kacprzak, Piotr Chomczyński, Jakub Ratajczak, Michał Marczak, Remigiusz Kozłowski, Dariusz Timler, Anna Pękala-Wojciechowska, Paweł Rasmus

**Affiliations:** 1Department of Medical Psychology, Faculty of Health Sciences, Medical University of Lodz, 90-419 Lodz, Poland; pawel.rasmus@umed.lodz.pl; 2Department of Applied Sociology and Social Work, Faculty of Economics and Sociology, University of Lodz, 90-136 Lodz, Poland; andrzej.kacprzak@uni.lodz.pl; 3Department of Sociology of Organization and Management, Faculty of Economics and Sociology, University of Lodz, 90-136 Lodz, Poland; piotr.chomczynski@uni.lodz.pl; 4Department of Management and Logistics in Healthcare, Medical University of Lodz, 90-419 Lodz, Poland; jakub.ratajczak@stud.umed.lodz.pl (J.R.); michal.marczak@umed.lodz.pl (M.M.); 5Center of Security Technologies in Logistics, Faculty of Management, University of Lodz, 90-136 Lodz, Poland; remigiusz.kozlowski@wz.uni.lodz.pl; 6Department of Emergency Medicine and Disaster Medicine, Medical University of Lodz, 90-419 Lodz, Poland; dariusz.timler@umed.lodz.pl; 7Department of Clinical Pharmacology, First Chair of Internal Diseases, Medical University of Lodz, 90-419 Lodz, Poland; anna.pekala-wojciechowska@umed.lodz.pl

**Keywords:** life course theory, criminal careers, qualitative research, juvenile offenders, adult offenders, public health

## Abstract

Both juvenile and adult criminal careers show regularities in the origins of delinquency, the dynamics of the criminal pathway, and the turning points that lead to desistance/persistence in crime. Research shows that family, education, and friendship environments contribute significantly to the individual choices that create criminal biographies. Our aim was to apply core aspects of life course theory (LCT): trajectory, the aged-graded process, transitions, institutions, and ultimately how desistance/persistence factor into explaining the criminal careers of Polish offenders. The research is based on in-depth interviews (130) carried out with both offenders (90) and experts (40). The offenders were divided into two groups: 30 were juveniles, and 60 were adults of whom half were sentenced for the first time (30) and half were recidivists (30) located in correctional institutions or released. The experts group (40) includes psychologists, educators, social rehabilitators, and prison and juvenile detention personnel working with offenders. We used triangulation of researcher, data, and methodology. Our data revealed that similar biographical experiences characterized by an early socialization, family and friends-based circles laid the groundwork for their entry and continued participation in criminal activity. Juvenile and adult first-time sentenced offenders led criminal careers significantly different from those of recidivists, who faced problems with social adaptation caused by lack of family and institutional support.

## 1. Introduction

Modern criminology is consequently moving towards cross-disciplinary research. One of its most influential research trends in recent decades has been life-course theory (LCT) related to the criminal career perspective, used in analyses of changes in individual criminal behavior over the course of a biography [1,2]. This approach is not a compact theory, it is rather eclectic in nature, bringing together representatives of different theoretical orientations. It includes elements of the concept derived from various trends, e.g., theories of social learning, social control, individual propensity theories or labeling approach [3]. A constitutive element of the criminal career perspective is the perception of an individual criminal career as a process composed of a number of sub-processes, the most important of which are: initiation of a criminal career (criminal onset), duration of criminal behaviors throughout life (persistence in crime) and the cessation of criminal activity (desistance from crime) [4,5,6].

A criminal career is a period of illegal activity of an individual, opened at a certain stage in one’s biography [7]. It is dynamic, processual and undergoes significant changes in the course of the biography [3]. These changes (transitions) may concern the profile of criminal activity (e.g., changing one’s criminal habits to committing burglaries instead of pickpocketing), the frequency of crimes committed (change in crime), periodic cessation of criminal activity (change in criminality) or desistance from crime [8]. Changes in criminal behavior in the course of a biography are a derivative of both endogenous factors (i.e., individual decisions, psychological predispositions, level of motivation to change and volitional activities) and exogenous factors (i.e., important life events in a person’s life, taking up non-criminal social roles, having the support of loved ones, experiencing acceptance in the environment, availability of public health and social care institutions) [1,4,6,8,9,10,11,12,13,14], lists eight most important correlations indicated by empirical analyses of criminal careers:The career usually begins between the age of 8 and 14;Early criminal initiation most often leads to a relatively long and “rich” career;The predominance of criminal behavior occurs in the late adolescence (15–19 years);Motives for crime up to the age of 19 are quite varied: for profit or “out of boredom” and more often aggressive, after the age of 20 economic motivation predominates;Most crimes before the age of 20 are committed in a group, later rather individually;Individual crime is usually part of a broader syndrome of anti-social behavior, the so-called antisocial versatility, including truancy, abuse of the weak, excessive drinking, drug consumption, reckless driving, irresponsible sexual behavior;A small fraction of the general offender population is responsible for a significant proportion of all crime—these offenders tend to start early in their careers, are more likely to commit crimes, and have longer careers than others;Desisting from crime most often takes place between the ages of 20 and 29.

There is a general consensus among criminologists that in most cases individual criminality (i.e., the intensity and frequency of offending) reaches its peak during late adolescence and early adulthood. It then declines steadily in later years. We find limited agreement on the reasons for this trend [15]. Some researchers have argued that the decline in criminal activity is related to the general decline in human vitality and activity resulting from the aging of the body. Glueck and Gottfredson [16,17] believe that major life events are of little relevance to an individual’s propensity to commit crime because it is determined by self-control, and crime decreases with age as opportunities to commit crime become fewer. However, Ezell and Cohen [18] reject this perspective, supporting Laub and Sampson [9] who argue that the retreat from delinquency is due to major life events, i.e., entering adulthood and fulfilling socially desirable social roles, which leads to a more conformist attitude.

Additionally, measuring the desisting from crime is particularly problematic. Researchers define it as the absence of recorded offences for several months after release from prison [19], a year with no recorded crimes [6], no arrests of juvenile offenders in adulthood [11] or not returning to prison within 10 years after release [20]. Nor is there complete agreement among researchers as to whether and in what situations a person can be said not to re-offend [21,22]. Some argue that there is a difference between definitive cessation of crime and periods of not committing any crime, highlighting the importance of identity change in this process. Edwin Lemert [23], argues that the periods of “silence” in a criminal career should be distinguished from the stage of complete “withdrawal” and acceptance of the identity of a “changed person”. Additionally, Laub and Sampson [24] emphasize the staged nature of the process, distinguishing termination—the period after which the individual no longer commits crimes, and desistance—a process leading to the cessation of criminal activity, involving a change of one’s identity, supported by the assumption of new social roles.

Research on the desistance from the crime process grew out of a critique of correctional institutions [25] and a focus on individuals who withdraw from crime without professional support, “spontaneous desisters” so to speak [26,27]. However, desistance from criminal activity can also be planned and achieved through projection [26]. Christy Visher [28] points out that programs supporting desistance from crime are particularly important in this process. The most widespread and effective programs are those aimed at increasing self-confidence, boosting self-esteem, seeing oneself as a person deserving to return to society, in particular educational programs, learning cognitive skills, psychological support and therapeutic interventions. Although Visher attributes an important role in the prevention of persistent crime also to instrumental support programs (e.g., finding a job and a place to live), in her opinion these are less effective. In contrast, the best results are achieved when both types of interventions are combined. The distinction between formal and informal support activities of ex-prisoners can be purely analytical. Offenders who “spontaneously” change their attitude towards crime seek support primarily from their relatives and friends. However, without professional help, they may have more difficulties, e.g., finding a job or maintaining abstinence. On the other hand, those who seek support from formal institutions and organizations also need support from their relatives. The most important thing is whether the support received brings the expected results, i.e., whether it contributes to the social reintegration of the offender [26].

## 2. Materials and Methods

Our research [29] took place between 2018 and 2020 and was funded by The Injured and Post-penitentiary Aid Fund of the Ministry of Justice (Grant of the Polish Ministry of Justice, No. DFS-II-7211-169/18/18 titled: ”Social determinants of juvenile and adult crime” (project coordinator: Piotr Chomczyński). Full report on this research project (in Polish) is available online [29]. There was a total of 130 participants in the study, of which 90 were juvenile (30) and adult (60) offenders. Half of the adult offenders’ group (30) were sentenced for the fists time and the rest (30) were recidivists. More than half of the offender population were serving sentences in prisons (55), the rest were already at large (35). Juvenile and adult offenders were convicted of various crimes (robbery, drug trafficking, theft, homicide, and physical assault). All of the offenders under study were Polish citizens coming from all over the country. The majority of them (73) were from lower class with poor educational and economic status while the rest (17) were middle class representatives. Juvenile offenders were at the age between 13 and 21 while adults were between 23 and 68. We obtained official access through formal negotiations with prison/jail authorities and liaison officers who helped us select prisoners who met our criteria [30]. The majority (28) of expert interviews (40) were conducted in both juvenile (correctional) and adult (correctional and remand) institutions. They worked as psychologists, social and mental health workers, rehabilitation specialists, vocational counsellors, educators, prison staff.

We used an open-ended biographical questionnaire designed to allow our interviewees to make spontaneous statements [31,32,33,34]. Initially, most of the inmates expressed reluctance to discuss their criminal experiences at the sight of the printed questionnaire, so we decided to use open-ended questions, similar to an informal conversation. Initially we thought this was an artefact of our outsider status. However, we found that their hesitation could be overcome by making our approach more interactive and participatory. We memorized categories and invited respondents to develop and co-create interview scenarios by adding their own questions or editing ours. This gave us a richer description than a formal interview tool, and our interviewees became less suspicious [30,34,35,36,37,38]. We believe that this method placed our subjects on a more egalitarian footing, because we were repeatedly invited back for additional in-depth interviews [30,33,39,40].

The duration of the interviews with offenders ranged from 30 min to 2 h, depending on the environmental conditions and the degree of trust placed in each respondent. The median age of the offenders was 27 years and the average age of onset of criminal activity was 17 years. The interviewers approached the subject of criminal careers carefully and made a conscious effort to create an atmosphere of openness and trust to obtain the most objective and detailed data possible. To facilitate this, they disclosed information about their personal biography and research objectives.

Triangulation of our data was particularly important as we were outsiders and dealing with a sensitive subject [35,41,42]. For that reason, we employed data, researcher, and methods triangulation.

We analyzed our data using ATLAS TI software (ATLAS.ti Scientific Software Development GmbH, Berlin, Germany) for qualitative data analyses, version 8.4.2. Key categories were inductively generated through open and selective coding of 130 in-depth interviews and illustrated with quotes presented in the paper that most closely reflected the conceptual framework discussed here [43,44,45]. To ensure anonymity and confidentiality, which are particularly needed when dealing with sensitive topics, all names used in our research are pseudonyms [30,39,46,47]. We edited the length and content of the quotations when necessary, to protect our subjects from being identified.

## 3. Results

### 3.1. Age-Graded Aspects of Transitions in Polish Offenders’ Criminal Careers

In 2020, the Polish police registered 786,302 crimes that were committed across the country (a decrease of 4.4% compared to 2019) and revealed 310,736 suspects, including 8287 juvenile perpetrators who were proven to have committed 18,951 criminal acts. Girls constituted a group of 1538 (18.6%) juvenile offenders [48]. According to Prison Service Statistics [49], at the end of 2020 there were 67,894 (3056 females) in correctional institutions, including juveniles aged 15–16 years (4), 17–18 years (146) and 19–21 years (1591). In the same year, the entire population of 384 juveniles (aged 13–21) was located in reformatories (307), detention centers (76) and around 6000 in youth educational centers (There are no official statistics on youths’ population in juvenile educational centers. The authors were informed about the population of youths located in reformatories and juvenile detention center by an informant from the Ministry of Justice because the statistics are not open-access available). Our aim is to go beyond statistics and undertake an in-depth biographical study of individual criminal careers from an LCT perspective. For this reason, we focus on the life stories of juvenile and adult offenders to verify which of the basic assumptions of the LCT apply to our results.

The empirical material indicates that the criminal onset is an indirect process, in which the state of social negligence plays a particularly significant role [50]. The socio-economic situation of the families from which our interviewees came formed a criminogenic background to their life choices. However, it should not be considered as a determinant factor for a deviant, and especially criminal future. The problem of child neglect becomes particularly significant when it occurs in in environments where behavioral and personality models and values contrary to generally accepted norms prevail. Hence, “poverty, unemployment, lack of life prospects and other manifestations of material exclusion are considered in criminology as genetic factors of crime” [51].

Patrick (age 17) illustrates the impact of an environment dominated by anti-social factors. Like many other adolescents he was exposed to the influence of incarcerated members of his family [52]. His early socialization and criminal experiences enabled him to define his “closed ones” prison experiences in terms of a taken for granted reality, as part of the surrounding objectivity [53,54] and that led to a subsequent prolific criminal career [4].

Researcher: Tell me, has any of your loved ones, apart from your dad, been punished, for example?Interviewee: Mom is being punished now.R: What is the reason?I: Insurance enforcement.R: Uh-huh, anyone else?I: All uncles, that is, three uncles.R: For what?I: For many reasons, the older they are the more crimes they commit.

The dominating narrative for the criminal onset is its placement in the context of multi-dimensional exclusion and neglect, which is often presented as a consequence of interviewees’ families’ lack of commitment to their upbringing and thus them “being raised by the street” [50]. Sampson and Laub [11] indicate that the process of criminal onset is conditioned by socialization content coming from various sources. It is therefore a complex combination of one’s attachment to conformist and non-conformist socializing milieus: family, peer groups and formal institutions such as school or social services [55,56]. Although Warr [57] tend to credit family as the main source of socially accepted norms and values in the process socialization, the biographies indicate that the family may transmit deviant socialization content as well.

Jan (35), who is a first-time sentenced adult with extensive experience of institutional care as a child, provides evidence of the negative impact of family on his later life. The lack of emotional security and support reinforced by physical punishment caused feelings of anxiety and there was little trust in his mother. She scrutinized his educational progress and herself beat him instead of supporting him. Jan’s early parenting problems multiplied his further failures and put a stigma on him as a troublemaker. This gradually excluded him from the circle of “good boys” [4,58,59,60].

I: Sometimes there were some outings that she (mother) was not at home. Caused by drinking alcohol. Sometimes it happened. It happened that she did not come home after 2–3 days.R: I understand that your family has had an alcohol problem because of excessive drinking. Father drank and froze. Mom was abusing too. How did this affect you and your brother? The fact that mom drank and disappeared for a couple of days?I: We sat at home and waited for her. The atmosphere was terrible then. We didn’t know if the mother would come back or not. This is how I learned to cook.R: I understand that, as you said, there was aggression in your home?I: Yes. Often also by the mother. When we were rude, we were beaten or when we had poor grades in school, we were also beaten.

Numerous studies confirm that early criminal initiation is a strong predictor of a long and prolific criminal career [4,9,61]. For the interviewed males who began their criminal careers as teenagers, it was particularly difficult to desist in the future. The research shows that juvenile criminalization and sentencing plays an important role as context to the formation of criminal identity which, in turn, constitutes a pivotal obstacle to one’s attempts to desist in the future.

Robert (46) is a repeat offender. From his biography we learn about his early start in crime. He started using drugs as a teenager and was involved in his first “petty thefts”. Drug consumption and first thefts decided about his smooth involvement into criminal career.

I: I tried drugs several times and I don’t like it […]R: What drugs?I: Amphetamine … Well, I mean, that’s when I started … when I stopped going to this school, problems started, petty thefts, I’ve been stealing car radios. They caught me and put to an emergency youth center, and it practically started … and then they locked me in a prison, and that’s how it went … no … everything …

Moreover, since the early emergence of crime, criminality was often the only known sphere of gainful activity. The abundance of a life of crime was associated with a corresponding lack of socially accepted roles. Early initiation was usually accompanied by dropping out of school structures, as well as the replacement of professional activity by routine criminal activity, which becomes the dominant or only form of income generation during the life course. This feature correlates with the “richness” of an early criminal career, i.e., with the high number and variety of crimes committed in the future [4]. These attitudes made it difficult to fulfil conformist social roles also in adult life, hence the social resources of the respondents were usually quite limited. Due to their involvement in a life of crime in their youth, they gave up education at its early stages, did not: gain professional qualifications, function on the labor market, have social capital in the form of friends not connected to the underworld. Thus, a situation of exclusion early in one’s life reflects and strongly influences later chances of desisting from crime. To solve the problem of social reintegration of ex-prisoners, it is therefore necessary to identify the sources and consequences of child neglect, as well as to continuously monitor the dynamics of the phenomenon.

LCT pays a lot of attention to turning points that are significant changes in the course of the biography [3,62,63] and may lead to positive or negative transitions [8]. Our interviewees’ narrations point at some family-based negative events that caused personal loss, feeling of abandonment, lack of support (parents’ divorce, increase of family violence or one of significant other death). Michał (19) points at his grandfather death at the turning point in his life. The experience of his relative pass away gave birth to gradual withdrawal from so far lifestyle along with alcohol and drug consumption that caused further problems with law.

I: I used to be a nice guy when I was younger, I was good, and then I stopped being nice.R: You stopped being nice and what happened?I: Generally, my grandfather died, and so I stopped to care about everything, and.R: Were you in a relationship with your grandfather?I: He was very … raised me … and so I started to go crazy a little.R: And what does it mean to go crazy?I: Well, there are cigarettes, alcohol, drugs.

Positive turning points usually lead to socially expected transitions in criminal career [6,8,9,10,11]. Key events, or “turning points” can trigger changes in an individual’s bond to society and hence in his/her engagement in crime [64].

Piotr (49) a repeat offender declares to find employment and re-build his relations with son.

I: and when I leave, I’ll be looking for a job, the first thing is the employment office, right? I will also file a letter for help with the court. Because in the beginning, I won’t have that money either. And I’ll be looking for a job, noR: I understand.I: and I’ll try to keep in touch with my son.

### 3.2. Criminal Careers, State-Dependence, and Public Health Issue

LCT pays a lot of attention to exogenous factors underlying desistance from crime (i.e., family, taking up non-criminal social roles, employment, availability of public health and welfare institutions) [6,9,10,11]. Our data show that a key determinant of social exclusion is the duration of imprisonment. Prisoners who have spent many years in prison, mainly repeat offenders, cannot rely on relatives and their chances for employment are much lower in comparison to juveniles and adults sentenced. For the first time. Julia the social worker gives example of typical problems faced by recidivists after their release, which may lead to persistence in crime.

R: I would like to ask about the problems that these people most often face.I: It is homelessness, no contact with families. Especially gentlemen who sat many times. Because they are mostly men. (E16-D, social worker)

Sampson and Laub [57] and also Warr [65] noted that: “Social institutions…may modify trajectories include school, work, the military, marriage, and parenthood.” Changes in an individual’s relationship with these various institutions are an inevitable feature of modern life and are therefore crucial to understanding desistance. Our research proves that lack of institutional support has great impact on offenders’ criminal careers and their action towards desistance. Marek (48) a family assistant and social worker points at insufficient inter-institutional cooperation as a factor that affects ex-offenders’ situation.

I: Theoretically there are some examples of institutional cooperation, but they are very formal. The sociotherapy center sends a letter a social assistance center to help someone in the form of refinancing or subsidizing food. The center sends a notification to the court that it would like the court to investigate a case. What drives a bureaucratic machine and has little to do with real aid.

A prisoner released from prison also faces an inability to function in outside environment. Our data reveal that the more years a person spends in prison, the less ready he/she is to take responsibility for his/her life. The social adaptation of ex-convicts requires some training aimed at gaining competences to function independently.

A person released from prison becomes helpless. Someone decides for him when to change the linen, when to have dinner, when to walk, and so on and so on. There is no institution that would take care of it.

## 4. Discussion

The research presented in this paper refers to the idea of social responsibility of science. We believe that the main scientific purpose and, parallelly, the most important social function of criminology is to support development of effective solutions aimed at crime prevention. This responsibility includes reducing the multidimensional negative consequences of crime experienced by at different levels: macrostructural (public expenditure on social services, e.g., health care or welfare), meso-structural (e.g., deterioration of neighborhoods with high crime rates) and microstructural (individual victimization and its impact on physical health and mental condition or ability to engage in social roles, etc.) [66,67,68,69,70,71,72,73,74]. Going behind statistics and identifying key determinants of criminal careers over the life cycle is a step towards constructing (more) effective crime prevention programs.

There is a further, fascinating way of possible implementation of the above research. It is the broadly understood individualized therapeutic approach: from psychotherapy to social work, from rehabilitation to vocational counselling and education in general. The link between data collection (explanatory function) and the potential practical use of this information to help reintegrate people caught up in criminal activity (predictive function) may be the approach itself. According to Daniel Bertaux [75], there are various ways in which life stories can be used in scientific and practical fields. It is said that even without a single standard methodology we can see them as a coherent point of view [76]. Moreover, there are already some studies confirming that such approach works [6]. The empowerment of this group will undoubtedly have a major impact on society. The limitation of this research comes from only social sciences methodology implementation (sociology, psychology, criminology). Life course criminology needs to go beyond “traditional” methods of data collection and reach for non-declarative tools (e.g., face and eye tracking) to deliver complementary and explanative information for developing so far knowledge.

A significant part of contemporary criminology is embedded in this paradigm of practical application. We observe this trend as new directions of research and analyses gain prominence and popularity among criminologists. One such direction, particularly cognitively interesting and promising in terms of practical application, is biosocial criminology. It attempts to incorporate an interdisciplinary (biological, sociological, and recently also psychological [77,78] evidence-based knowledge into the practical field of work aiming at social rehabilitation of persons committing crime throughout the life-course. The complexity of crime—in terms of its multidimensional consequences—requires a multifaceted analysis carried out jointly by representatives of various scientific disciplines [79,80]. Simultaneously, there is a growing recognition of the need to address individuality in projecting and applying crime prevention and social reintegration programs [81].

## 5. Conclusions

The life-course approach provides an important background for understanding individual pathways to initiation and persistence in crime. Furthermore, and perhaps even more importantly, it allows us to better understand individual responsiveness to prevention interventions and thus provides us with more detailed understanding of the factors that contribute to individual desistance from crime. We believe that the future of crime prevention programing, including through offenders’ social rehabilitation lies in the research that develops around this approach.

## Data Availability

Non-digital data supporting this study are curated at Medical University of Lodz.

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
