# Peer review of "Age-Graded Transitions and Turning Points in Polish Offenders’ Criminal Careers from the Standpoint of Life Course Theory"

_ijerph, 2021, doi:10.3390/ijerph18116010_

Round 1

Reviewer 1 Report

Summary: The paper details a research on trajectory, the aged-graded process, transitions, institutions, and ultimately 25 how desistance/persistence in explaining criminal careers of Polish offenders using LCT. The subject of the paper, methodology, and the findings are all very important to the intended population of the journal. The idea of going beyond the numbers and looking into the determinants of criminal careers over their life cycle is intriguing. The methods, interviews, and discussions are worthy content to be published. Overall, the paper is well written and is ready for publication with minor English check.

Comments/Questions:

  • The idea of biosocial criminology is introduced later in the paper, but this could be done in the introduction to better set up the work.
  • It is not clear what open-ended questions were used during the interviews.

Originality/Novelty: The methods and the results are, to my knowledge, original and well defined.

Quality of Presentation: The writing is comprehensible and appropriate.

Interest to the Readers: The work is of interest to the readership of the journal.

Overall Merit: The work contributes to the field and has benefits. I suggest publishing after minor English check. 

English: Needs minor improvement. I suggest proofreading the document again for typos, grammar, and use of prepositions. For example, see typo on line 48, line 125, etc.

Author Response

Response to Reviewer 1 Comments

Dear Sir/Madam,

        We wish to express our appreciation for the comments and suggestions for our manuscript entitled “Age-graded transitions and turning points in polish offenders’ criminal careers from the standpoint of life course theory”. We have carefully revised the manuscript taking into consideration all the comments.

  • The idea of biosocial criminology is introduced later in the paper, but this could be done in the introduction to better set up the work.

The aim of this paper was to focus on explaining criminal careers of Polish offenders using Life Course Theory. Therefore, we decided to signalize this subject in the discussion as loosely related to the topic.

  • It is not clear what open-ended questions were used during the interviews.

As we mentioned in the text: “We used an open-ended biographical questionnaire designed to allow our interviewees to make spontaneous statements [31-34]. Initially, most of the inmates expressed reluctance to discuss their criminal experiences at the sight of the printed questionnaire, so we decided to use open-ended questions, similar to an informal conversation.”

  • Originality/Novelty: The methods and the results are, to my knowledge, original and well defined.

Thank you.

  • Quality of Presentation: The writing is comprehensible and appropriate.

Thank you.

  • Interest to the Readers: The work is of interest to the readership of the journal.

Thank you.

  • Overall Merit: The work contributes to the field and has benefits. I suggest publishing after minor English check. 

Thank you. We have scanned the text for any typos, grammar, and use of prepositions and have corrected the mistakes.

  • English: Needs minor improvement. I suggest proofreading the document again for typos, grammar, and use of prepositions. For example, see typo on line 48, line 125, etc.

We have corrected the mistakes.

Once again we would like to sincerely thank you for a very comprehensive, insightful and in many places accurate review. All of the above comments are highly valuable for the comprehensiveness of the paper and our own scientific development.

Regards

Reviewer 2 Report

Overview:  The population is a bit unclear as to who made up the 130 participants and who was interviewed.  Additional information would be helpful in clarifying the population studied. 

Abstract:  Thorough and concise with good explanation of population, purpose, and findings

Introduction: 

Line 48 – there is a 4 in the middle of the word “proce4sses”

Materials and Methods:

The number of prisoners was disclosed but more specific demographics would be helpful.  The author mentions that there were 130 participants which consisted of 90 juvenile and adult offenders and out of those,  30 were repeat offenders.  Please clarify if the 30 repeats were part of the juvenile and adult offender group or if they were separate.  If the 130 participants were not juvenile or adult offenders, what were they (what accounts for the other 40 participants)?  Do you have descriptions of exact age ranges or ethnic/race diversity? Are there any other descriptive data that could be included regarding the descriptions of participants?  Socioeconomic status?  Geographic location? If not, why?

Please clarify if the interviews conducted were with professional mental health workers and staff or with the prisoners.  This is a bit confusing. 

Lines 135-144 – this is good clarification as to how the methods were adjusted to prioritize openness of responses. 

Line 147-149 – the description of the inmates – does this apply only to the interviews or to the entire study of 130 participants?

Results:

Lines 310-316 – This is a very clear description of the underlying causes of issues when released from prison.

Discussion:

Lines 338-345 – with recent movements in the psychological field towards the biopsychosocial model, it is also important in criminology so it is nice to see it presented here. 

Conclusion:

The conclusion is good for not only the results, but also supporting the use of the methodology.

Author Response

Response to Reviewer 2 Comments

Dear Sir/Madam,

        We wish to express our appreciation for the comments and suggestions for our manuscript entitled “Age-graded transitions and turning points in polish offenders’ criminal careers from the standpoint of life course theory”. We have carefully revised the manuscript taking into consideration all the comments.

  • Overview:  The population is a bit unclear as to who made up the 130 participants and who was interviewed.  Additional information would be helpful in clarifying the population studied. 

We clarified information concerning the group under study and enriched it with additional data. The corrections are visible both in Abstract, Material and methods.

  • Abstract:  Thorough and concise with good explanation of population, purpose, and findings

Thank you.

Introduction: 

  • Line 48 – there is a 4 in the middle of the word “proce4sses”

We corrected this mistake.

Materials and Methods:

  • The number of prisoners was disclosed but more specific demographics would be helpful.  The author mentions that there were 130 participants which consisted of 90 juvenile and adult offenders and out of those,  30 were repeat offenders.  Please clarify if the 30 repeats were part of the juvenile and adult offender group or if they were separate.  If the 130 participants were not juvenile or adult offenders, what were they (what accounts for the other 40 participants)?  Do you have descriptions of exact age ranges or ethnic/race diversity? Are there any other descriptive data that could be included regarding the descriptions of participants?  Socioeconomic status?  Geographic location? If not, why?
  • Please clarify if the interviews conducted were with professional mental health workers and staff or with the prisoners.  This is a bit confusing. 

We clarified this information. It can be seen in Abstract and Materials and methods sections.

  • Lines 135-144 – this is good clarification as to how the methods were adjusted to prioritize openness of responses. 

Thank you.

  • Line 147-149 – the description of the inmates – does this apply only to the interviews or to the entire study of 130 participants?

We specified that it applies only to offenders.

Results:

  • Lines 310-316 – This is a very clear description of the underlying causes of issues when released from prison.

Thank you.

Discussion:

  • Lines 338-345 – with recent movements in the psychological field towards the biopsychosocial model, it is also important in criminology so it is nice to see it presented here. 

Thank you.

Conclusion:

  • The conclusion is good for not only the results, but also supporting the use of the methodology.

Thank you.

Once again we would like to sincerely thank you for a very comprehensive, insightful and in many places accurate review. All of the above comments are highly valuable for the comprehensiveness of the paper and our own scientific development.

Regards

Reviewer 3 Report

  1.  

    1. It is unnecessary to add page numbers in the citation. Please move the page information to the references.
    2. I suggest authors list all the questionnaires as supplements of this paper?
    3. In 3.1, it is better to calculate the percentage for each data as well.
    4. Line 157 in page 4, what is the software for? Please explain it in detail.
    5. In the material section, it is quite important to describe the background of the study area.
    6. As a qualitative study, I do not think the analysis results are representativeness of many places. It is a trend that authors use more data and apply quantitative methods (i.e., text mining) to explore the patterns.
    7. In the discussion, I suggest authors have more discussions about the findings of study, limitations and future work.

Author Response

Response to Reviewer 3 Comments

Dear Sir/Madam,

        We wish to express our appreciation for the comments and suggestions for our manuscript entitled “age-graded transitions and turning points in polish offenders’ criminal careers from the standpoint of life course theory”. We have carefully revised the manuscript taking into consideration all the comments.

1. It is unnecessary to add page numbers in the citation. Please move the page information to the references.

We moved the page numbers to the references and removed them from citations.

2. I suggest authors list all the questionnaires as supplements of this paper?

In qualitative study like ours it is unlikely to add used tools because of their length and cultural/language barriers. Translation from Polish language to English would change the questions accuracy. Moreover, we used four different tools to each group of research participants (experts, juveniles, first-time sentenced and recidivists).

3. In 3.1, it is better to calculate the percentage for each data as well.

These data are presented like this by the authors who we cite, therefore we are not sure if recalculating them would be correct. Statistics we used are non-evoked data for us and part of desk research.

4. Line 157 in page 4, what is the software for? Please explain it in detail.

We put additional information concerning the purpose of using enlisted software.

5. In the material section, it is quite important to describe the background of the study area.

The study was based on organizational ethnography approach. We argued that: “We obtained official access through formal negotiations with prison/jail authorities and liaison officers who helped us select prisoners who met our criteria [30]. The majority (28) of expert interviews (40) were conducted in both juvenile (correctional) and adult (correctional and remand) institutions. They worked as psychologists, social and mental health workers, rehabilitation specialists, vocational counsellors, educators, prison staff.”

6. As a qualitative study, I do not think the analysis results are representativeness of many places. It is a trend that authors use more data and apply quantitative methods (i.e., text mining) to explore the patterns.

Applying qualitative approach, we focused on delivering in-depth and exploratory insight in criminal careers of juvenile and adult offenders. Our aim was to understand nuances lying under choices made by offenders that took a form of regularities contributing to criminal trajectories. Despite statistical representatives we employed theoretical saturation and theoretical sampling to develop better understanding of family, environmental and peers’ influence on criminal onset and persistence due to LCT assumptions.

7. In the discussion, I suggest authors have more discussions about the findings of study, limitations and future work.

We added additional information covering findings of this study and its limitations and possible future work in Discussion section.

Once again we would like to sincerely thank you for a very comprehensive, insightful and in many places accurate review. All of the above comments are highly valuable for the comprehensiveness of the paper and our own scientific development.

Regards